# A Deep-Learning Model with Task-Specific Bounding Box Regressors and Conditional Back-Propagation for Moving Object Detection in ADAS Applications

**DOI:** 10.3390/s20185269

**Published:** 2020-09-15

**Authors:** Guan-Ting Lin, Vinay Malligere Shivanna, Jiun-In Guo

**Affiliations:** Department of Electronics Engineering and Institute of Electronics, National Chiao Tung University, Hsinchu 30010, Taiwan; victor.ep01@g2.nctu.edu.tw (G.-T.L.); jiguo@nctu.edu.tw (J.-I.G.)

**Keywords:** conditional back propagation, convolutional neural network, road objects detection

## Abstract

This paper proposes a deep-learning model with task-specific bounding box regressors (TSBBRs) and conditional back-propagation mechanisms for detection of objects in motion for advanced driver assistance system (ADAS) applications. The proposed model separates the object detection networks for objects of different sizes and applies the proposed algorithm to achieve better detection results for both larger and tinier objects. For larger objects, a neural network with a larger visual receptive field is used to acquire information from larger areas. For the detection of tinier objects, the network of a smaller receptive field utilizes fine grain features. A conditional back-propagation mechanism yields different types of TSBBRs to perform data-driven learning for the set criterion and learn the representation of different object sizes without degrading each other. The design of dual-path object bounding box regressors can simultaneously detect objects in various kinds of dissimilar scales and aspect ratios. Only a single inference of neural network is needed for each frame to support the detection of multiple types of object, such as bicycles, motorbikes, cars, buses, trucks, and pedestrians, and to locate their exact positions. The proposed model was developed and implemented on different NVIDIA devices such as 1080 Ti, DRIVE-PX2 and Jetson TX-2 with the respective processing performance of 67 frames per second (fps), 19.4 fps, and 8.9 fps for the video input of 448 × 448 resolution, respectively. The proposed model can detect objects as small as 13 × 13 pixels and achieves 86.54% accuracy on a publicly available Pascal Visual Object Class (VOC) car database and 82.4% mean average precision (mAP) on a large collection of common road real scenes database (iVS database).

## 1. Introduction

In recent years, deep-learning algorithms have contributed to huge advancements in the development of self-driving cars. By learning from abundant and peculiar databases, the neural networks can realize the rules hidden in the black box. All that the researchers have to do is to design an efficient architecture and make it learn well. For the visual perception of vehicles, the convolutional neural networks (CNNs) are one of the most powerful types of architecture [1,2,3,4,5]. Various vision applications such as object detection, object recognition, semantic segmentation and so on are based on these. Among the aforementioned, the object detection is considered most significant and preferred task for autonomous driving [6,7,8,9,10] vehicles.

Object-detection algorithms face the challenges of simultaneously detecting different objects of diverse scaling sizes as shown in Figure 1. This also demonstrates the complexity of detecting objects in the real driving scenarios. For the vehicles driven at higher speed on the highways and freeways, detecting obstacles distanced far from the vehicle becomes a critical issue of this application. For instance, consider a real situation of a car travelling at 120 km/h; it can be mathematically established that the car can come into collision with an object 100 m away in 3 s.

Drivers who do not fully concentrate typically need about 1–2 s [11] to react to a sudden event without any driver assistance system leaving behind only 2 s to prevent the probable collision with the objects leading to major casualties. In the case of cities, heavy traffic also leads to the distraction of drivers making it the most common cause of car accidents. To make it applicable for all kinds of driving environment, the detection algorithms should be capable of locating objects of different scales ranging from tinier to larger simultaneously and precisely. For real-world applications, reducing the computational cost is a key designing concern. The object detection algorithms nowadays often only either focus on accuracy [12,13,14,15] or computational speed [16,17,18] perspectives. For the accuracy perspective, predicting the location of tiny instances is cumbersome as they often appear with blurred boundaries and implicit appearances making it hard to distinguish them from their background. As far as larger objects are concerned, these typically exhibit severe differences of visual representation. As a result, two-stage detectors [12,13] are designed to extract a large amount of information from images to classify all the objects of varying scales. Although larger configurations provide affluent information for recognition, it makes the whole system slow and memory-intensive. If speed is weighed up as the primary design concern, one-stage detectors [17,18] dismiss the complex pipeline in two-stage detectors, and predict the location and object type at the same time. This simplification makes the networks work faster but reduces the capability of detecting objects of different scales, especially smaller objects.

Motivated by both one-stage and two-stage detectors, this paper proposes novel task-specific bounding box regressors (TSBBRs), built based on a one-stage detector pipeline. The proposed TSBBRs shown in Figure 2 are capable of fetching and decoding specific objects independently. As shown in Figure 2, given an input image with arbitrary scales of objects in it, the features of whole image would be extracted by the backbone layers. Then, treating the extracted features from the backbone as input, two TSBBRs fetch and decode specific information and generate target objects’ predictions. Finally, all predictions would be merged and passed to non-maximum-suppression (NMS) procedure.

With the aim of empowering the detection module to recognize the task-specific features from inputs, a conditional back-propagation mechanism that makes each module have its own learning target and update the weights at the same time is developed in the proposed algorithm. 

The proposed conditional back-propagation mechanism is discussed in detail in Section 3.5. It is in the criterion of the short side of bounding boxes. The back-propagation would be performed only if it meets the experimentally preset criterion. Instead of alternative training for TSBBRs, the training flow is made end to end. The loss would be collected and summed up with weighing in single forward passing. After forward passing, the weights in the networks would be updated simultaneously. This not only simplifies the training procedure but also makes the networks a better solution for such applications. In our experiments, it is found that using the short side of the bounding boxes as a back-propagation criterion is the most effective way to merge the TSBBRs.

The novelty and contributions of this paper can be listed as below:(1)First, novel and modular TSBBRs for detecting the objects of all scales in the real-time scenarios on the road are proposed and implemented. The proposed models can run at 67 fps on an experimental setup equipped with NVIDIA 1080TI GPU, at 19.4 fps on an autonomous vehicle platform i.e., NVIDIA Drive PX2, and at 8.9 fps on NVIDIA embedded Jetson TX2.(2)Secondly, to make TSBBRs learn their own targets without degradation; a conditional back-propagation mechanism to update the weights in different TSBBRs simultaneously to reach the global minimum is developed.

## 2. Related Work

CNNs [1,2,3] have been a sensation in the research field of computer vision. They provide a way to extract powerful visual models that yield hierarchies of features. A well-designed network through end-to-end training even surpassed the human ability of visual perception [19]. 

Object detection tasks are some of the key applications of the advanced driver assistance systems (ADAS) and self-driving cars. The vital requirement of such systems is that it can detect the salient objects as precise as possible, which means that the algorithm should eliminate the false inference during testing time and raise the recall rate. Detecting objects distanced from as far as possible is also an important key to this technology. Another application issue is that the algorithm should perform efficiently in real time under the restricted memory capacity and computational power on a portable platform.

The state-of-the-art methods of object detection can be broadly classified into methods based on two-stage detectors and one-stage detectors.

### 2.1. Computer Vision-Based Methods

Previously, vehicle detection was based on computer vision (CV) methods. The process of car detection was discriminated depending on contour information. The representative methods based on non-contour information are those that used histogram-based features (HOG, etc.) [8,20,21], Haar-like features [22,23] and the implicit shape model (ISM) [24]. Villamizar et al. acquired local statistical information using a simple comparison between two bins of Histogram of Oriented Gradients (HOG) [21]. Ozuysal et al. built a Scale-Invariant Feature Transform (SIFT)-histogram pyramid for car detection at multiple scales [20]. A Haar-like strip feature was employed to learn the structure of cars [23]. The Implicit Shape Models (ISM) is a widely used technique in object detection where generalized Hough transform is successfully applied to localize target objects. In contrast, other researches that use either contour or edge based features include [25,26,27]. In [25], the whole idea of ISM was recaptured using part-based contour fragments as features followed by an attempt to improve the recognition accuracy using oriented chamfer matching. However, it seemed to be sensitive to shifting and rotation variations. On the other hand, the variation of object appearances was learned using a deformable feature called active contour fragment (ACF) [26] and a middle-level feature called ‘Sketch Tokens’ [27]. There was also an effort to use heterogeneous features including edgelet, HOG and covariance rather than employing a single feature [28]. As for object localization, Hough transform-based center voting schemes were popularly exploited.

With the change in the research trends in recent years and the big leap towards the CNN-based methods for object detection, this paper has proposed a CNN-based method for detecting cars in the real traffic environment.

### 2.2. Two-Stage Detectors

The region-based object detectors with convolutional neural networks [12,13,29,30,31,32,33,34] split the detection pipeline into a region proposal stage and a classification stage. Faster-Region-CNN (RCNN) and its descending methods utilize the convolutional-based configuration to produce region proposals accurately, called region proposal network (RPN).

RPNs find out that the convolutional features can also be used for generating region proposals. RPN is constructed on top of the features extracted by the backbone network by adding additional convolutional layers that can perform the regression of bounding boxes and object scores at each location. The proposal regions would be sent to a classifier to determine the object type. RPN can be trained according to the type of application for generating detection proposals. Hong et al. [35] proposes a deep but lightweight architecture with Concatenated Rectified Linear Unit (C. ReLU) and inception modules that improve accuracy and reduce computational cost. Based on the Faster-RCNN pipeline, Lin et al. [13] produces region proposals with different levels of feature maps in the network.

### 2.3. One-Stage Detectors

One stage detectors [17,18,36,37,38] are an analogy of the human perception system. When people view images, they immediately know the objects in the image, where they are and how they interact with each other. The human visual system is fast and accurate, allowing us to perform complex tasks, such as driving cars or locating the objects without much conscious thought. A fast and accurate target detection algorithm allows the computer to drive the cars without a dedicated sensor, enabling the auxiliary device to transmit real-time scene information to the user and dig the potential of generic purpose in response to the robot system. 

You Only Look Once (YOLO) and its descending elements [16,18] reframe object detection pipeline into a single regression problem, which encodes image pixels to bounding box coordinates and class probabilities at one inference. It divides the input image into an S × S grid. The grid cell is responsible for detecting the object if the center of an object falls into this grid cell. A single CNN simultaneously predicts multiple bounding boxes and class probabilities for each box. YOLO [16,18] is trained on entire images and optimized for detection performance. This unified model has several benefits compared to the regional-based methods of object detection. YOLO is [16,18] extremely fast compared to Faster-RCNN. Since it frames detection as a regression problem, no complex pipeline is needed for the system. To simply run CNN on an image predict detections for testing, YOLO runs at 45fps on a modern graphics processing unit (GPU) device and the fast version runs at more than 150fps, which means that it can process streaming video in real-time specification with less than 30 milliseconds of latency. Besides the improvement of speed, YOLOv2 [18] reaches comparable mean average precision (mAP) with the Faster-RCNN system on publicly available database [39]. Although YOLO [16,18] provides an efficient way for inferencing, its single path design makes it hard to detect various scales of objects, especially small objects.

## 3. Proposed Model with Task-Specific Bounding Box Regressors (TSBBRs) and Conditional Back-Propagation Mechanism 

### 3.1. Overview of the Proposed Model

This paper proposes an accurate and uncomplicated convolutional neural network for simultaneously detecting various objects of different scales. With a dual-path convolutional network design and conditional back-propagation, the proposed architecture can detect various scales of common object on the roads in real-time and it is robust to different driving environments with a single camera input.

As shown in Figure 2, the proposed algorithm can be divided into three parts namely, backbone feature extractor, object detection modules, and NMS. Initially, an input image would be passed through a simple pre-processing procedure, which subtracts every pixel with global mean values. The global mean values are calculated pixel wise from the large-scale image classification database, ImageNet-1 k [40]. Then, a less complex, shared computation feature extraction backbone follows. It encodes the features of an image into a generic and compact representation followed by a task-specific sub-module performing object detection in parallel. Finally, all detection results are processed with NMS to eliminate highly overlapped boxes.

In this algorithm, the ratio of the short side and its corresponding image dimension of an object are used to distinguish between a tiny object from that of a larger one. It is empirically found that when setting the criterion near the reciprocal of the dimension of network output of detecting tinier objects, the model can be made to converge easily which in the given method is 1/14 = 0.0714. The impact of this criterion is briefly described in Section 3.5.

### 3.2. Mechanisms of Bounding Box Regressors

Since our focus is object detection in the mobile systems that require low power consumption and have restricted memory capacities, one-stage detectors are employed in the proposed method instead of regional based object detectors in order to reduce the complexity. One-stage detection systems like YOLO [16] take the detection as a regression problem. The entire image is fed into the network. On top of the extracted features produced by the backbone, the bounding box regressors (BBRs) predict the object’s bounding boxes in the image by dividing the image into multiple grids and each predicts the bounding box co-ordinates with *x* and *y* as object center and *w*, *h* denoting its spatial dimensions, objectness prior, and vector that contains class probabilities. If there were no object center located within the grid box, the objectness prior would be set as 0 and otherwise set as 1. The final object scores *P* in grid box (*i*, *j*) would be calculated using Equation (1)
(1)$$P\left(i,j\right)=\max\left(p_{ij}\;\times C_{ij}\right)$$
where *p_ij_* is the objectness prior of the grid box (*i*, *j*) and *C_ij_* is the vector with probabilities of each class.

### 3.3. Bounding Box Regressors with Large Receptive Field

The YOLO [16] system splits the input image into 8×8  grid boxes in order to detect the larger objects. The graphical demonstration of this case in a real driving environment is as shown in Figure 3a. The grid annotated in red is an example of the grid containing larger objects and the yellow grid denotes the smaller objects. It can be observed that when the objects are small and located near to each other, one grid box can be filled with more than one object as in the yellow grid. This results in detection degradation when there are many tiny objects in the training database. The BBRs permit the detection of only one object in each grid box, which would make ambiguous back-propagation in the training stage. For example, in Figure 3a, the object B in the yellow box would be treated as background because the object A has a slightly larger bounding box compared to the object B. However, in human perception, the feature of the object B is undoubtedly a car. As a result, the model would produce an imprecise and low confidence score for these types of objects, which in turn leads to lower recall rate and mAP in the detection stage.

### 3.4. Bounding Box Regressor with Small Receptive Field

To remedy the defect of BBRs with large receptive field, this paper up-scales the number of grid boxes by nearly two times, as shown in Figure 3b, which increases grid boxes from 8×8 to 14×14. The smaller objects are now relocated to different grids. This prevents the ambiguous back-propagation for tiny objects in the training stage since each grid box has its own propagation target. However, naively scaling up the number of grid box raises another issue. The BBRs with 14 × 14 grid boxes are numerically unstable in the training phase. The loss spread within an unreasonable range implies that it lowers the network performance compared to employing 8×8 grid boxes. It is experimentally found that this phenomenon is the impact of detecting the larger objects. In Figure 3b, the object cropped in the red box crosses the boundaries of four grid boxes, which means that each grid box contains part of the information of this object. As BBR only back-propagates for the box in which the center of an object resides, grid box A is the only one treated as a positive instance. The grid boxes B, C and D will be regarded as background no matter how much an object is spread in it. This causes another ambiguous issue because the network forces the confidence scores of grid boxes near the center, but not located at 0.

### 3.5. Conditional Back-Propagation Mechanism

The setting of different numbers of grid boxes has its pros and cons; BBRs set with a larger number of grid boxes enhance the performance by detecting the smaller objects but encounter numerical instability due to the ambiguous back-propagation. By contrast, although BBRs set to a smaller number of grid boxes are numerically more stable, they encounter difficulty while detecting smaller objects. Since the goal of this work is to meet the requirement of detecting all kinds of objects in the driving environments, this algorithm tries to explore the advantages of different kinds of BBR settings and try to get rid of the negative impacts. 

To detect various scales of object at the same time, one probable solution is to separate the small and large objects in the database, and then train the corresponding configuration of architectures independently. This can prevent the ambiguous back-propagation and produce high accuracy bounding boxes. However, running different detection modules separately causes intensive computation and memory cost during both training and testing stages. In the training stage, the divided database takes at least double the disk space to store the separated data and each set of weights is to be trained separately as well, making the training procedure tedious and time-consuming. On the other hand, it takes more than twice the usual memory cost in the testing stage. All models perform forward passing independently without sharing computation. The restricted memory size and bandwidth make the parallelism of models almost impossible when it is realized on mobile systems. Since our goal is to make the algorithm run on the systems that can be mounted on unrestrained vehicles, separating models will not be an ideal design choice. It is also needed to reduce the complexity of the proposed system as far as possible. 

To incorporate different specialties of module in a unified framework, this paper proposes a conditional back-propagation mechanism as shown Figure 4. In the training and testing stages, the computations of the backbone layers are shared among different settings of BBRs. The BBRs are designed in a task-specific manner, that is, each BBR has its own target to detect according to the previously described criterion. These kinds of BBRs are called task-specific bounding box regressors (TSBBRs).

The training of TSBBRs optimizes the following multi-part loss function as shown in Equation (2).
(2)$$C_{coord}\overset{I}{\underset{i=0}{\sum }}\overset{J}{\underset{j=0}{\sum }}O_{ij}\left[{\left(x_{ij}-{\hat{x}}_{ij}\right)}^{2}+{\left(y_{ij}-{\hat{y}}_{ij}\right)}^{2}\right]\;\;+C_{coord}\overset{I}{\underset{i=0}{\sum }}\overset{J}{\underset{j=0}{\sum }}O_{ij}\left[{\left(w_{ij}-{\hat{w}}_{ij}\right)}^{2}+{\left(h_{ij}-{\hat{h}}_{i}\right)}^{2}\right]\;+C_{obj}\overset{I}{\underset{i=0}{\sum }}\overset{J}{\underset{j=0}{\sum }}O_{ij}\left[{\left(P_{ij}-{\hat{P}}_{ij}\right)}^{2}\right]\;+C_{noobj}\overset{I}{\underset{i=0}{\sum }}\overset{J}{\underset{j=0}{\sum }}\left(1-O_{ij}\right)\left[{\left(P_{ij}-{\hat{P}}_{ij}\right)}^{2}\right]\;+\overset{I}{\underset{i=0}{\sum }}\overset{J}{\underset{j=0}{\sum }}O_{ij}\|(C_{ij}-{\hat{C}}_{ij}\|{}_{2}$$
where *C_coord_*, *C_obj_* and *C_noobj_* denotes the weighing factors of coordination and objectness prior scores, (*x*, *y*, *w*, *h*) is spatial dimension of bounding box, *P_ij_* is objectness prior, *C_ij_* is an hot vector that contains class probabilities, and *O_ij_* = 1 if the center of object in matched criterion appears in cell (*i*, *j*) and 0 otherwise. 

The conditional back-propagation mechanism is in the criterion of the short side of the bounding boxes. If the ratio of the short side of an object and its corresponding image side is less than the pre-set criterion mentioned in 0, it is considered as a tiny object, otherwise, as a large object. The objects that fail to meet the criterion of TSBBR would be seen as background in the forward-propagation stage. In the proposed model, two TSBBRs are used to detect different scales of objects concerning accuracy and efficiency. The TSBBR for tiny objects has a smaller receptive field. It splits the input image into 14×14  grid boxes. On the other hand, the TSBBR for large objects is configured with a larger receptive field that splits the given image into 8×8  grid boxes. It should be noted that this loss function penalizes the classification error only if a center of the matched object is located in that grid cell and it only penalizes the coordinate error of bounding boxes if the matched object center is located inside. In practical terms, the empty grids dominate the number of cells in an image. As the result, the confidence scores of grid boxes tend to be 0 for the over powering of gradient from the empty boxes. We set *C_coord_* = 5, *C_noobj_* = 0.5 and *C_obj_* = 5 in both TSBBRs. In the end-to-end training procedure, the balance of the loss function would also affect the model performance. To collect the overall loss of the model in forward-passing, it is experimentally found by setting the weighing loss for small TSBBRs twice that of the large ones, which results in the most stable training progress.

### 3.6. Configuration of TSBBRs

The configuration details of the backbone layer is as shown in Figure 5 and the corresponding details along with both TSBBRs used in the proposed algorithm are listed in Table 1, Table 2 and Table 3, respectively. The TSBBRs consist of two parts. The first part is to fetch and encode the task-relative information from features extracted by backbone layers. The second part is convolutional kernels for the prediction of each grid box. The encoding part is built with stacking 3×3 convolutional kernels in depth of 1024. In order to shield the computation and enrich the solution space, S×S convolutional kernels with a depth of 512 are inserted. The influence of the value of *K* is discussed in Section 4. The last convolutional layer with a size of 3×3 in a depth of B×5+C plays the role of information decoder. It performs regression operation for predicting bounding boxes information. The parameter B indicates the number of boxes per grid that should be predicted and C is the number of classes. In the iVS database comprising of cars, pedestrians and motorbikes illustrated in Appendix A, C is set to 3 and for the Pascal VOC 2007 car [39] database, C is set to 1. For tiny objects, B = 3, and for large objects B = 5 are employed based on the experiments conducted and to which the best performance is observed. In addition, the batch normalization [41] for convergence and regularization consideration is used and PReLU [19] as the activation function in both TSBBRs.

### 3.7. Training of Deep Network

The network in the proposed algorithm is trained for about 120 epochs on the training split of the objective database. The Pascal VOC 2007 Car database consists of around 1963 samples where the iVS database has around 1383 samples. The datasets are randomly split in the ratio of 4:1 implying 80% of the data are used for training and the remaining 20% are used for testing. Throughout the training, a batch size of 100, a momentum of 0.9 and a decay of 0.0005 is employed. The learning procedure scheduled is as follows: For the first epoch, the learning rate of 10^−3^ is used. If it is initialized with large learning rates, unstable gradients make the trained model diverge. The model training with the learning rate 10^−2^ for 70 epochs is continued followed by using the learning rate 10^−3^ to train for 30 epochs, and finally, the learning rate 10^−4^ is employed to train for 20 epochs. To enhance the robustness of the trained model, a run-time data augmentation technique is applied during training. For data augmentation, a customized layer to perform random color transformation, dropout, cropping and rotation for all databases is implemented. The samples are color transformed in the range of [−20%, +20%] in the respective red, green and blue (RGB) channels, rotated randomly in range of [−30°, +30°] and randomly cropped in the range of 70–100% of the original image and at random object locations as shown in Figure 6. All the training and inference is built upon the Caffe framework [42].

### 3.8. Inference

The prediction on an input image only requires single forward passing. In the iVS database, the network predicts 833 bounding boxes per image and class probabilities for each grid box. Figure 2 shows the proposed system’s flow of detection. The proposed model is fast during testing process since it only requires a single network passing unlike regional based methods. The computation cost of backbone layers is shared for both TSBBRs and task-specific encoders pass forward in parallel. The input images would be resized to the size of 448 × 448 and subtracted with global mean value calculated from the ImageNet-1 k [40] database. The large objects or the objects near the border of multiple cells would be localized well in multiple grid boxes. Finally, the non-maximum suppression for all boxes predicted from both TSBBRs at same time to acquire the ‘real-maximum’ boxes is performed.

## 4. Discussion

### 4.1. Ablation Experiments

All the ablation experiments and analysis on the iVS database are conducted in order to evaluate the significance of each of the designing factors. The mAP, AP and IoU are all calculated as per [43]. Average precision (AP) is the most commonly used metric to measure the accuracy of object detection by various CNN, and image-processing methods. The AP computes the average precision value for recall value. Precision measures how accurate the predictions are, by a method, i.e., the percentage of correct predictions whereas, Recall measures the extent to which the predicted positives are good. Equation (3) is employed in this paper to estimate the AP where *r* refers to recall rate and $\hat{r}$ refers to the precision value for recall. The interpolated average precision [44] was used to evaluate both classification and detection. The intention in interpolating the precision/recall curve in this way is to reduce the impact of the wiggles in the precision/recall values, caused by small variations in the ranking. Similarly, the mean average precision (mAP) is the average of AP.
(3)$$AP=\sum (r_{n+1}\;-\;r_{n})\;p_{interp}\;(r_{n+1})\;p_{interp}(r_{n}{+}_{1})\;=\underset{\hat{r}\geq r_{n+1}}{max}p\left(\hat{r}\right)$$

#### 4.1.1. Impact of the Backbone

To explore the impact of feature extraction networks, a single BBR is trained and tested with different backbone layers. For fair comparisons, exactly the same configuration of each BBR is applied. The images are split into 8 × 8 grids and each grid predicts 5 boxes (B = 5) at the same time. 

The experimental results are listed in Table 4 and Table 5. It proves the fact that more powerful the backbone layers, higher the mAP. The trade-off for the mAP increase is the computational cost. However, by replacing the GoogleNet with the powerful ResNet-152, only a marginal improvement in the mAP is achieved but there is 5 × longer forward time. This shows that the performance of the backbone layers does matter in the final detection quality whereas, the gain will be saturated as the ability and complexity go up. For accuracy and speed considerations, DarkNet-19 is employed in the proposed backbone layers for the remaining experiments.

#### 4.1.2. Overlapping Criterion in Conditional Back-Propagation

To train the TSBBRs, it is necessary to set the prior criterion of each TSBBR for the conditional back propagation mechanism. It is found that when faceted value is set as criterion by the reciprocal of output dimension of 1/14 = 0.0714 the tiny object network, the model fails to detect the objects with the size near the criterion boundary. To overcome this issue, the back-propagation criterion in the proposed method is slightly overlapped. By extensive experiments, the upper bound criterion of 0.08 to the BBR for tiny objects and lower bound of 0.07 for larger one are chosen. For extreme cases, a lower bound of 1.0 for tiny objects and 0.0 for the larger objects are chosen and the model diverges in training time as discussed in Section 3.

#### 4.1.3. Shielding Computation Layers of TSBBRs 

In Table 1 and Table 2, one of the convolutional kernels with half the number of filters between the original layers is inserted. This not only reduces the theoretical cost of computation between large convolutional kernels but also provides a richer solution spaces of weights. In this paper, overlapping criterion described in a previous study for the experiment is used. Ideally, *K* = 0 (no shielding layers) and *K* = 3 cases in Table 1 and Table 2 should have the same computational costs, but when the overhead of data synchronization of previous layers is considered, shielding computation layers still increase the forward time by about 6.7% and mAP by 5.2%, as seen in Table 6. 

#### 4.1.4. Pass through

In the field of convolutional neural networks for computer vision, it is a common practice to fuse the features of lower levels into the higher [5,45], for global information in the higher resolves what while local information in the lower resolves where. The width and height of the feature maps produced by the 13th convolutional layer is reshaped into the size of the 19th layer in Darknet-19 followed by the concatenation of them along the depth dimension. Although low-level features contain richer spatial information, they have some redundant information [46,47] at the same time. From our experiments, it is found that naive concatenation leads to degraded result as in the first row of Table 4. It can be argued that this comes from the fully connected specialty via channel-wise direction of 2-D convolutional operation; the convolutional kernel can be approximated as a channel-wise fully-connected layer, so it would be harmful if all the concatenated information is adopted for classification or regression. To simultaneously utilize the spatial and semantic information, a 1×1 locally connected convolution layer such as grouped convolutions [48,49,50] is used to sort out the unwanted features. The weights in this kernel dynamically tune the importance that is the value of its connected features during the training stage. The configurations of TSBBRs with pass through design are shown in Table 7 and Table 8. The results are shown in Table 9. It is experimentally determined that when the concatenated information is divided into 8 groups, the best performance is achieved. The mAP increases when the number of groups increase from 1 to 8. However, when the input features are further split into 16 groups, the mAP decreases because the overly-sparse features would make it hard for TSBBRs to encode the semantic and spatial information of the bounding boxes.

## 5. Results

The effectiveness of the proposed TSBBRs and conditional back-propagation mechanism is evaluated on the iVS database and publicly available Pascal VOC 2007 car [39] database. Furthermore, CarSim [51] simulation software is also used to validate the model’s ability of detecting the vehicles at faraway distances. The end of the section lists a comparison of the performances of our model on the portable devices to validate the efficiency of the proposed system.

### 5.1. Performance Comparison on the iVS Database

Some of the state-of-the-art models along with their corresponding datasets on which they are trained, tested, and respective comparison with the proposed method are as in Table 10. In order to compare the proposed model with that of the other state-of-the-art models, the other real-time object detectors were trained with the iVS database. The experiments are repeated with the released procedure of training on the respective official websites. The results of the same are as shown in Table 11 and Table 12. 

Besides YOLOv3 [36], the proposed model with *K* = 1 and *K* = 3 outperform other one-stage detectors as the ability to detect tiny objects still remains efficient. Even though there is a marginal mAP reduction of 2.4% compared to YOLOv3 [36], the inference of our models requires only 60% of the computation cost in the testing stage. The timing results are measured on NVIDIA 1080TI GPU device.

### 5.2. Experiments on Pascal VOC 2007 Car Database

The proposed algorithm is evaluated on the publicly available Pascal VOC 2007 car dataset [39] Based on the descriptions in [6], 1434 images containing cars in the *trainval* and test sets in the Pascal VOC 2007 dataset are extracted to evaluate the proposed method. For fair comparison, the same exact evaluation criterion of Pascal evaluation metrics with intersection over union (IoU) set as 0.7 to assess the correctness of localization mentioned in [6,7] is employed. For data augmentation, besides the aforementioned strategies, more data augmentation methods when training on the Pascal VOC 2007 car *trainval* split are used. Moreover, to enhance the robustness of both TSBBR for tiny and the large objects on such small dataset, the input frames are randomly resized into a smaller scale and patched on a three-channel canvas initialized with random noises in the original size to make the proposed architecture invariant to scales and noises. Table 13 tabulates the experimental results on the Pascal VOC 2007 dataset, where the information is partially adopted from [6,7]. The models of [7] are trained on an even larger-scale database called CompCars [52]. This contains more than 100,000 web-nature data as the positive training samples that are annotated with tight 2-D bounding-boxes. The proposed model still achieves a better result even on limited amount of training data. To strive for a better performance, we also compare the model that is initially pre-trained on the iVS database and then fine-tuned on the Pascal 2007 car *trainval* split. With this initialization procedure, the performance is boosted by almost 20% from 66.83% to 86.54% in AP perspective, as shown in Table 11. Figure 7 shows some of the results of the proposed algorithm on the Pascal VOC 2007 car dataset. Our method detects the cars that are occluded, located near the border and those hidden within other vehicles that are in analogous appearance.

### 5.3. Evaluation of Far Distance Object Detection on Simulation Software

Simulation software is commonly used in quantitative experiments as it provides accurate and quantitatively available mutual information between vehicles and environment without manual labeling. In this paper, CarSim [51] simulation software is used to test the maximum detection distance of the proposed models. The proposed model (*K* = 3, pass through, group = 8) is compared with YOLOv2, both trained on the iVS database. As shown in Figure 6, the proposed model is able to detect the car from a distance 124 m away while the counterpart can only detect the object within 45 m if the detected box that has IoU > 0.5 of the ground is regarded as the true prediction. However, if the IoU criterion is set to 0.3, the objects at 200 m away with the short side object of 13 pixels can be detected. Figure 8 shows the results of the object detection by proposed algorithm on the CarSim simulation software.

### 5.4. Performance Evaluation on Portable Devices

Besides accuracy, the efficiency of the proposed method is also crucial in order to be employed in real-world applications. To verify the performance of the proposed mechanism, it is implemented and tested on low-power mobile platforms. The two platforms chosen for verification are NVIDIA Drive PX2 and Jetson TX2. NVIDIA Drive-PX2 platform is a preferred system for ADAS applications due to its low power consumption and its processing efficiency. The Jetson TX2 is a low power platform for embedded systems designed to support multi-core processing and GPU acceleration. The quantitative results of efficiency on these two platforms are listed in Table 14.

## 6. Conclusions and Future Works

This paper proposed an efficient multiple path fully convolutional neural network to detect objects of various scales, ranging from tinier to larger objects, simultaneously. The proposed conditional back propagation mechanism makes each detection module learn without any performance degradation. The proposed shielding computation design and data augmentation also improves the robustness of BBRs (bounding box regressors). The proposed multiple path network design yields mAP of 82.4% on the iVS database and 86.54% mAP on a combined publicly available Pascal VOC database and iVS database. The proposed design can detect an object with roughly 13 pixels in the short side, at a distance 200 m away on CarSim simulation software. In the future, additional object types can be added in the proposed design to provide more object-detection information for self-driving cars.

The proposed method can be extended to detect other objects in real traffic environments such as traffic signs, road-dividers, road humps and other obstacles on the road such as rocks that have fallen on the road, roads dug up for maintenance works, and so on. Additionally, a further analysis can be carried to investigate a furthermore efficient network architecture that is also compatible with low-power and restricted memory applications. 

## Figures and Tables

**Figure 1 sensors-20-05269-f001:**
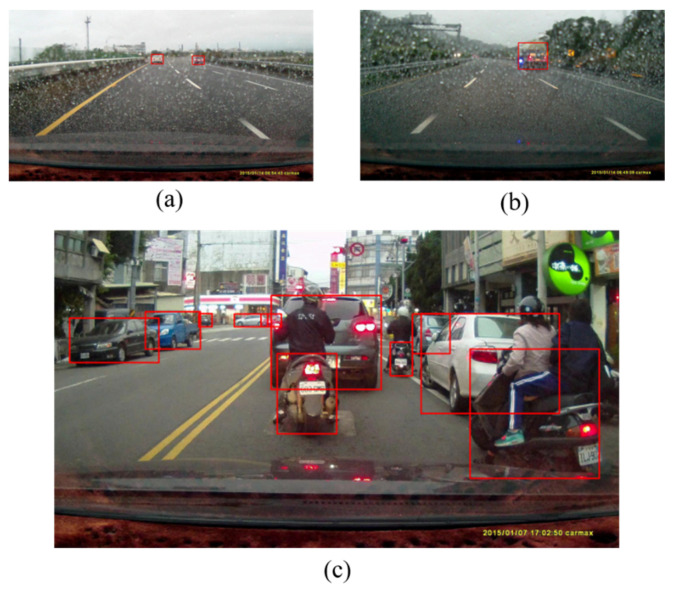
Illustration of the complexity of real driving environments. In real driving environments, objects often appear in different kinds of scale and distance as in (**a**–**c**).

**Figure 2 sensors-20-05269-f002:**
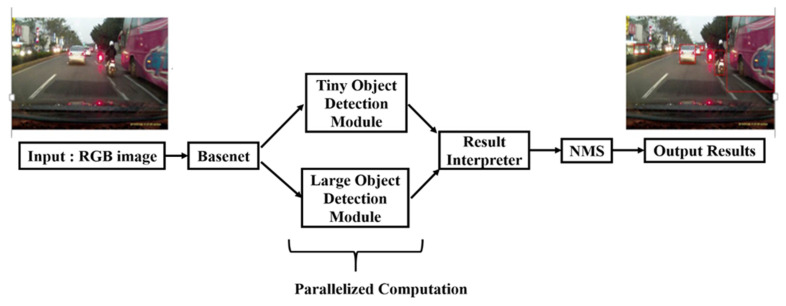
Object detection system with proposed task-specific bounding box regressors (TSBBRs). Given a color image as input, general features are extracted by the shared backbone. With these features as input, TSBBRs fetch and decode their own related information followed by coordination and class prediction simultaneously. Finally, all predicted boxes would be merged and passed into the non-maximum-suppression (NMS) procedure to eliminate the redundant boxes.

**Figure 3 sensors-20-05269-f003:**
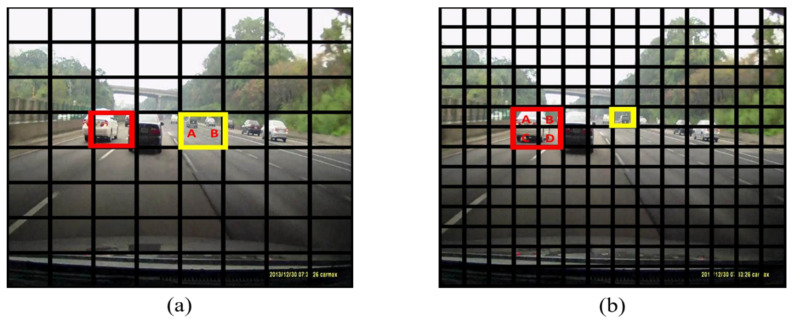
Visual analysis of different settings of bounding box regressors (BBRs). (**a**) Tiny objects (A and B) would locate in the same grid box. Ambiguous back-propagation results in lower recall rate. (**b**) Large objects (A, B, C and D) that cross the boundary of many grid boxes, makes a numerical unstable phenomenon during the training stage.

**Figure 4 sensors-20-05269-f004:**
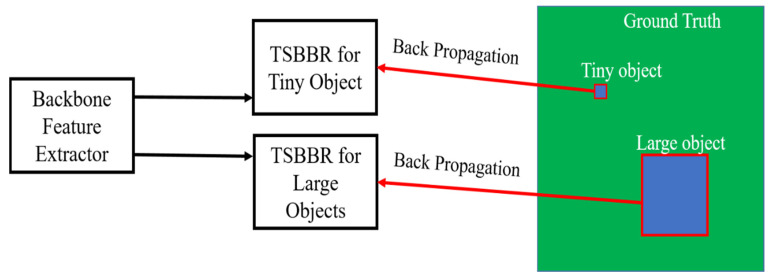
Proposed conditional back-propagation mechanism. According to the ratio of the object short side and corresponding frame side of the ground-truth, TSBBRs would only produce losses on their prior set targets. These losses would be collected and summed up. Based on the summed losses, weights in model would be updated simultaneously.

**Figure 5 sensors-20-05269-f005:**
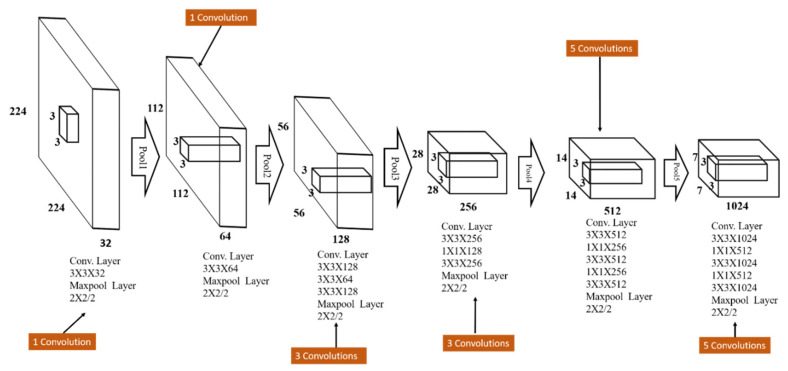
The architecture of the proposed TSBBRs network consisting of 18 convolution layers.

**Figure 6 sensors-20-05269-f006:**
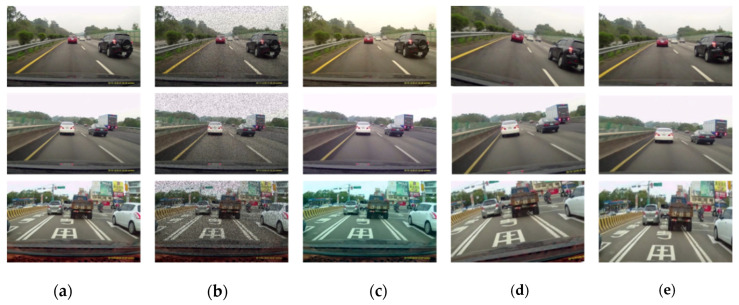
Examples of augmented data. (**a**) Original images, (**b**) dropout, (**c**) color transformation, (**d**) rotation, (**e**) cropping.

**Figure 7 sensors-20-05269-f007:**
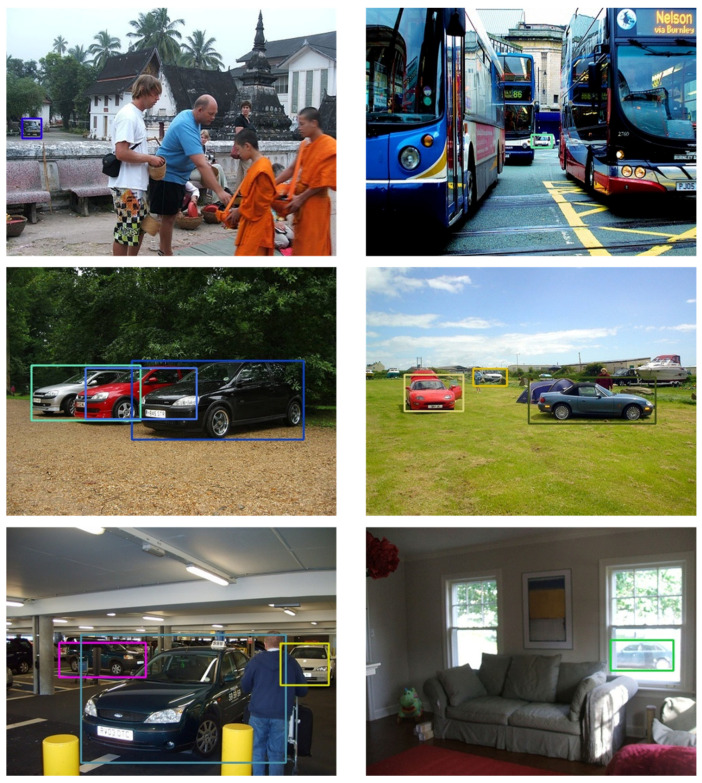
Detection examples on Pascal VOC 2007 car database. Our model can detect the cars that are occluded, located near the border or hidden within other vehicles that are of analogous appearance.

**Figure 8 sensors-20-05269-f008:**
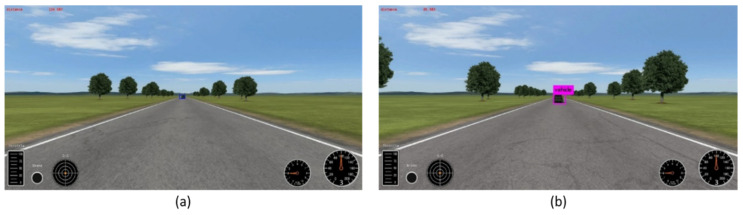
Detection results on CarSim simulation software. (**a**) the result of the proposed model, detecting a vehicle 124 m (intersection over union (IoU) > 0.5) and 200 m (IoU > 0.3) away. (**b**) the result of YOLOv2, detecting the vehicle within 45 m (IoU > 0.5).

**Table 1 sensors-20-05269-t001:** Configuration of backbone layers.

Type	Filters	Size/Stride	Output
Convolutional	32	3 × 3	224 × 224
Max-pooling		2 × 2/2	112 × 112
Convolutional	64	3 × 3	112 × 112
Max-pooling		2 × 2/2	56 × 56
Convolutional	128	3 × 3	56 × 56
Convolutional	64	1 × 1	56 × 56
Convolutional	128	3 × 3	56 × 56
Max-pooling		2 × 2/2	28 × 28
Convolutional	256	3 × 3	28 × 28
Convolutional	128	1 × 1	28 × 28
Convolutional	256	3 × 3	28 × 28
Max-pooling		2 × 2/2	14 × 14
Convolutional	512	3 × 3	14 × 14
Convolutional	256	1 x 1	14 × 14
Convolutional	512	3 × 3	14 × 14
Convolutional	256	1 × 1	14 × 14
Convolutional	512	3 × 3	14 × 14
Max-pooling		2 × 2/2	7 × 7
Convolutional	1024	3 × 3	7 × 7
Convolutional	512	1 × 1	7 × 7
Convolutional	1024	3 × 3	7 × 7
Convolutional	512	1 × 1	7 × 7
Convolutional	1024	3 × 3	7 × 7

**Table 2 sensors-20-05269-t002:** Configuration of TSBBR for tiny objects.

Type	Filters	Size/Stride	Output
Convolutional	1024	3 × 3	14 × 14
Convolutional	512	S × S	14 × 14
Convolutional	1024	3 × 3	14 × 14
Convolutional	512	S × S	14 × 14
Convolutional	1024	3 × 3	14 × 14
Convolutional	18	3 × 3	14 × 14

**Table 3 sensors-20-05269-t003:** Configuration of TSBBR for large objects.

Type	Filters	Size/Stride	Output
Max-pooling		2 × 2/2	7 × 7
Convolutional	1024	3 × 3	7 × 7
Convolutional	512	S × S	7 × 7
Convolutional	1024	3 × 3	7 × 7
Convolutional	512	S × S	7 × 7
Convolutional	1024	3 × 3	7 × 7
Convolutional	28	3 × 3	7 × 7

**Table 4 sensors-20-05269-t004:** The mean average precision (mAP) of different backbone layers.

Model	mAP (%)	Vehicle AP (%)	Bike AP (%)	Pedestrian AP (%)	FPS
Alexnet + BBR	28.0	37.7	36.7	9.5	330.9
GoogleNet + BBR	48.6	69.5	65.8	10.4	94.3
Darknet-19 + BBR	51.9	61.7	36.1	57.7	92.3
ResNet-152 + BBR	52.1	62.1	39.6	54.7	18.6

**Table 5 sensors-20-05269-t005:** The mAP of different overlapping criterion of conditional back-propagation mechanism, where **nov** denotes the non-overlapping case, **ov** denotes the overlapping case, and **ext** denotes the extreme case.

Model	mAP (%)	Vehicle AP (%)	Bike AP (%)	Pedestrian AP (%)	FPS
TSBBRs-nov	72.8	81.0	75.5	61.8	66.7
TSBBRs-ov	75.6	81.9	75.9	68.9	
TSBBRs-ext	N/A	N/A	N/A	N/A	

**Table 6 sensors-20-05269-t006:** Results of different settings of shielding computation.

Model	mAP (%)	Vehicle AP (%)	Bike AP (%)	Pedestrian AP (%)	FPS
*K* = 0	75.6	81.9	75.9	68.9	66.7
*K* = 1	77.1	84.2	77.0	70.0	71.9
*K* = 3	80.8	86.0	74.4	81.9	62.5

**Table 7 sensors-20-05269-t007:** Configuration of pass through design for tiny objects.

Type	Filters	Size/Stride	Output
Group Convolutional	512	1 × 1	14 × 14
Convolutional	1024	3 × 3	14 × 14
Convolutional	512	1 × 1	14 × 14
Convolutional	1024	3 × 3	14 × 14
Convolutional	512	1 × 1	14 × 14
Convolutional	1024	3 × 3	14 × 14
Convolutional	18	3 × 3	14 × 14

**Table 8 sensors-20-05269-t008:** Configuration of pass through design for large objects.

Type	Filters	Size/Stride	Output
Group Convolutional	512	1 × 1	7 × 7
Max-pooling	512	2 × 2/2	7 × 7
Convolutional	1024	3 × 3	7 × 7
Convolutional	512	1 × 1	7 × 7
Convolutional	1024	3 × 3	7 × 7
Convolutional	512	1 × 1	7 × 7
Convolutional	1024	3 × 3	7 × 7
Convolutional	28	3 × 3	7 × 7

**Table 9 sensors-20-05269-t009:** The mAP of different groups of pass through.

Number of Groups	mAP (%)	Vehicle AP (%)	Bike AP (%)	Pedestrian AP (%)	FPS
1	77.2	82.3	76.1	73.2	64.1
4	78.4	86.5	76.5	72.2	67.1
8	82.4	86.8	78.5	82.0	67.3
16	73.8	83.8	72.4	59.2	66.0

**Table 10 sensors-20-05269-t010:** Comparison of the proposed method with the state-of-the-art methods.

Models	Input	Dataset(s)	FPS	mAP (%)
YOLOv2 416 [18]	416 × 416	Imagenet, COCO	67	76.8
YOLOv2 608 [18]	608 × 608	Imagenet, COCO	40	78.6
YOLOv3 [36]	320 × 320	Open Images Dataset, COCO	X	28.2
SSD 300 [17]	300 × 300	PASCAL VOC, COCO, ILSVRC	59	74.3
SSD 512 [17]	512 × 512	PASCAL VOC, COCO, ILSVRC	59	76.9
Proposed TSBBRs *K* = 1, pass through (group = 8)	448 × 448	PASCAL VOC, iVS Database	67.03	77.1
Proposed TSBBRs *K* = 3, pass through (group = 8)	448 × 448	PASCAL VOC, iVS Database	67.03	80.8

**Table 11 sensors-20-05269-t011:** Comparison of computational cost.

Models	Multiply and Accumulate (MAC) (G/frame)	FPS
YOLOv2 416 [18]	16.56	68.5
YOLOv2 608 [18]	30.49	37.4
YOLOv3 [36]	28.12	35.0
SSD 300 [17]	30.53	46.0
SSD 512 [17]	87.84	19.0
Proposed TSBBRs *K* = 1, pass through (group = 8)	14.83	78.2
Proposed TSBBRs *K* = 3, pass through (group = 8)	16.89	67.3

**Table 12 sensors-20-05269-t012:** Comparison of the proposed design to other models on the iVS database.

Models	mAP (%)	Vehicle AP (%)	Bike AP (%)	Pedestrian AP (%)
YOLOv2 416 [18]	69.6	68.2	74.1	66.6
YOLOv2 608 [18]	77.9	80.7	73.4	79.7
YOLOv3 [36]	84.8	86.8	80.3	87.1
SSD 300 [17]	65.1	77.3	73.1	44.9
SSD 512 [17]	79.2	93.8	72.5	71.3
Proposed TSBBRs *K* = 1, pass through (group = 8)	79.0	86.8	78.5	71.8
Proposed TSBBRs *K* = 3, pass through (group = 8)	82.4	86.6	78.5	82.0

**Table 13 sensors-20-05269-t013:** Experiment results of the Pascal VOC 2007 car dataset.

Methods	Training Data	Average Precision (AP)	Processing Speed (FPS)
RCNN [30]	VOC07 car trainval	38.52 %	0.08
Fast RCNN [32]	VOC07 car trainval	52.95 %	0.5
Faster RCNN [32]	VOC07 car trainval	59.82 %	6
RV-CNN [6]	VOC07 car trainval	63.91 %	-
FVPN [48]	CompCars dataset [52]	65.12 %	46
Proposed (TSBBRs *K* = 3, pass through, group = 8)	VOC07 car trainval	66.83 %	67.3
Proposed (TSBBRs *K* = 3, pass through, group = 8)	The iVS database + VOC07 car trainval	86.54 %	67.3

**Table 14 sensors-20-05269-t014:** Performance of the proposed design on two portable platforms.

Model	Platform	Frames Per Second (FPS)
Proposed model (TSBBRs *K* = 3, pass through, group = 8)	NVIDIA Drive-PX2	19.4
	NVIDIA Jetson TX2	8.9
YOLOv3	NVIDIA Drive-PX2	8.5
	NVIDIA Jetson TX2	3.2

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
