# Peer review of "A Deep-Learning Model with Task-Specific Bounding Box Regressors and Conditional Back-Propagation for Moving Object Detection in ADAS Applications"

_sensors, 2020, doi:10.3390/s20185269_

Round 1

Reviewer 1 Report

 This work proposes a deep learning system that exploits Task-Specific Bounding Box Regressors (TSBBRs) and Conditional Back-Propagation mechanisms in automotive applications to recognize tiny and large objects in separate way.

The paper is well-written and organized, the proposed experiments appear valid and exhaustive and the results worthy of note but there are some issues to fix:

1) The related work section must be expanded: for each of the suggested references must be provided the final results, the datasets used and the neural network models exploited (backbone and so on..). You can use tables to summarize such information.

2) In Section 3.6 a scheme of the network overall architecture must be provided, highlighting each section (backbone, etc..) and how they are linked together (middle layers, etc..) specially when implement the Conditional back-propagation mechanism.

3) Section 3.7 must be expanded: a description of the dataset and references to previous works where it has been used and how it has been made is needed.

4) In section 3.7 splitting procedure description is needed (which are classes and their samples number? is there a K-fold cross validation?). Moreover, data augmentation layer must be described for reproducibility.

5) Since in the experiments there are empiric parameter values (both TSBBRs, in setting the reciprocal of the dimension of network output of detecting tinier objects (pag.4), the last convolutional layer of TSBBR) it would be better to summarize all them in a new Section, highlighting how much they can vary or become complicated when changing test datasets or network models.

Minor issues:

Abstract: define ADAS

Pag.2: “typically need about 1 second” -> put a reference

Pag.3: “… as shown in Figure 3.” -> Figure 3 must be placed where first mentioned. Moreover, it is not clear if the paragraph refers instead to figure 4 for the system architecture compared with figure 2. In this way, Figure 2 and 4 seems very basic, the systems can be better summarized with schemes including inputs and outputs for every stage, together with the features and the tools exploited.

Pag. 3: “When people despise images, they…” -> this sentence is not clear or lacks references

Pag. 4: “… as shown in Figure 2,” -> is figure 4 ?

Pag.4: In paragraph 3.1 there is an error for a section reference.  

Pag.4: “…values are calculated channel wise ..” -> this sentence is not clear, define better the approach

Pag.5: the description of the cases in Figure 3 is not very clear, moreover you can use different letters in 3(b) from 3(a)

Pag.5: “which increases grid boxes from 7x7” -> in 3(a) seems that there are 8x8 grid boxes 

Section 4.1: the evaluation metrics should be defined (mAP, FPS, IoU, etc ..)

Page 11: “Table tabulates the experimental results …” -> lacks the table number

Author Response

Dear Reviewer-1,

We appreciate your comments and we have revised the manuscript and responded to your comments as in the attached file.

Thank you.

-Authors

TSBBRS

Reviewer 2 Report

-Please correct it in your paper
Error! Reference source not found..

-please format text and formulas it is not written clear.
-please add block diagram of the proposed research;;;;
-please add photo of measurements of the proposed research + arrows what is what;;;;
-please add photo of application of the proposed research. Maybe for factories, refineries?
-please add some sentences about future analysis;;;
-What is formula for Vehicle AP [%], AP [%]
-Please cite references from Web of Science 2018-2020, please show new knowledge.

Author Response

Dear Reviewer-2,

We appreciate your comments and we have revised the manuscript and responded to your comments as in the attached file.

Thank you.

-Authors

TSBBRS

Reviewer 3 Report

In this paper, a deep learning model with task-specific bounding box regressors (TSBBR) and conditional back-propagation method is proposed to detect moving objects in ADAS applications. The proposed method can both detect larger and tinier objects, e.g. cars, in an ADAS application. In the proposed method, the authors applied Darknet-19 + BBR as the backbone layer. In the computational results, in iVS database, the proposed method achieves highest FPS than YOLOv2, YOLOv3, and SSD. Although YOLO v3 has better mAP, Vehicle AP, Bike AP and Pedestrian AP than the proposed method for 0.2%-5%, but the proposed FPS is almost twice than YOLOv3. It demonstrates that this method is proper to be used in real-time applications. On the other hand, for Pascal VOC2007 car dataset, the proposed method achieves highest average precision and FPS than RCNN, FVPN methods. Finally, the proposed method achieves 19.4 FPS on a portable platform but YOLOv3 only 8.5 FPS. The results in this paper can motive the possible readers in this research field. However, there are some still points need to be revised:

1. The sentences in the second and third paragraphs in Sec. 1 need some refrences. For example: “…a car travelling at 120km/hr…collision…100 meters…in 3 seconds”, “…about 1 second to react…behind only 2 seconds…”  

2. In Figure 2, there are two “Tiny object detection module” blocks. The authors should rename one of the block’s name and add more descriptions.

3. In the sentences in the last second paragraph of Sec. 1, may the correct Figure 3 is missing. Please add a flowchart to help readers for better understanding for the sentences in the paragraph.

4. The last sentence of Sec. 3.1 is missing. “Error Reference source not found”. Please correct this issue.

Author Response

Dear Reviewer-3,

We appreciate your comments and we have revised the manuscript and responded to your comments as in the attached file.

Thank you.

-Authors

TSBBRs

Round 2

Reviewer 1 Report

Authors responded quite satisfactorily to the doubts expressed above, I think the paper is ready for publication.

Author Response

Dear Reviewer-1,

We, the authors of sensors-836713, sincerely thank the reviewer for the recommendation of acceptance.

Thank you.

-Authors,

Sensors-836713

Reviewer 2 Report

-please add arrows to figures (what is what)

-Tables what is AP, formula (3). Please describe it.

-Experiments should be described better.

-compare methods with other techniques of image processing

for example 

-Recognition of images of finger skin with application of histogram, image filtration and K-NN classifier

-Automatic shape recognition of film sequence with application of backpropagation neural network

Author Response

Dear Reviewer-2,

We, the authors of sensors-836713, sincerely thank you for your comments to improve our manuscript. We have revised the manuscript accordingly. 

Thank you.

-Authors,

Sensors-836713

Reviewer 3 Report

My raised points are well addressed and modified in this version of the paper. I suggest this paper to be accepted.

Author Response

Dear Reviewer-3,

We, the authors of sensors-836713, sincerely thank the reviewer for the recommendation of acceptance.

Thank you.

-Authors,

Sensors-836713